# DCN+: Mixed objective and deep residual coattention for question answering

**Caiming Xiong,**\* **Victor Zhong,**\* **Richard Socher**
Salesforce Research
Palo Alto, CA 94301, USA
{cxiong, vzhong, rsocher}@salesforce.com

## ABSTRACT

Traditional models for question answering optimize using cross entropy loss, which encourages exact answers at the cost of penalizing nearby or overlapping answers that are sometimes equally accurate. We propose a mixed objective that combines cross entropy loss with self-critical policy learning. The objective uses rewards derived from word overlap to solve the misalignment between evaluation metric and optimization objective. In addition to the mixed objective, we improve dynamic coattention networks (DCN) with a deep residual coattention encoder that is inspired by recent work in deep self-attention and residual networks. Our proposals improve model performance across question types and input lengths, especially for long questions that requires the ability to capture long-term dependencies. On the Stanford Question Answering Dataset, our model achieves state-of-the-art results with 75.1% exact match accuracy and 83.1% F1, while the ensemble obtains 78.9% exact match accuracy and 86.0% F1.

## 1 INTRODUCTION

Existing state-of-the-art question answering models are trained to produce exact answer spans for a question and a document. In this setting, a ground truth answer used to supervise the model is defined as a start and an end position within the document. Existing training approaches optimize using cross entropy loss over the two positions. However, this suffers from a fundamental disconnect between the optimization, which is tied to the position of a particular ground truth answer span, and the evaluation, which is based on the textual content of the answer. This disconnect is especially harmful in cases where answers that are textually similar to, but distinct in positions from, the ground truth are penalized in the same fashion as answers that are textually dissimilar. For example, suppose we are given the sentence "Some believe that the Golden State Warriors team of 2017 is one of the greatest teams in NBA history", the question "which team is considered to be one of the greatest teams in NBA history", and a ground truth answer of "the Golden State Warriors team of 2017". The span "Warriors" is also a correct answer, but from the perspective of traditional cross entropy based training it is no better than the span "history".

To address this problem, we propose a mixed objective that combines traditional cross entropy loss over positions with a measure of word overlap trained with reinforcement learning. We obtain the latter objective using self-critical policy learning in which the reward is based on word overlap between the proposed answer and the ground truth answer. Our mixed objective brings two benefits: (i) the reinforcement learning objective encourages answers that are textually similar to the ground truth answer and discourages those that are not; (ii) the cross entropy objective significantly facilitates policy learning by encouraging trajectories that are known to be correct. The resulting objective is one that is both faithful to the evaluation metric and converges quickly in practice.

In addition to our mixed training objective, we extend the Dynamic Coattention Network (DCN) by Xiong et al. (2017) with a deep residual coattention encoder. This allows the network to build richer representations of the input by enabling each input sequence to attend to previous attention contexts. Vaswani et al. (2017) show that the stacking of attention layers helps model long-range

---

\*Equal contribution

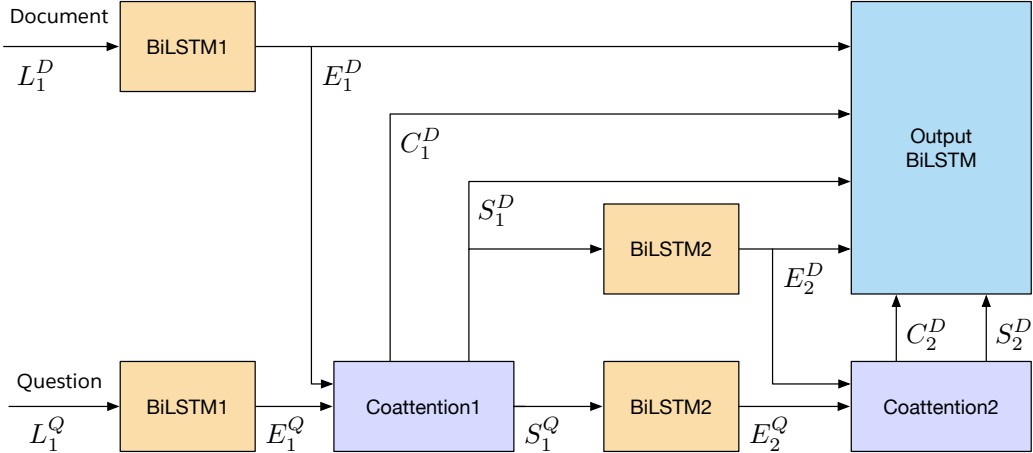

Figure 1: Deep residual coattention encoder.

dependencies. We merge coattention outputs from each layer by means of residual connections to reduce the length of signal paths. He et al. (2016) show that skip layer connections facilitate signal propagation and alleviate gradient degradation.

The combination of the deep residual coattention encoder and the mixed objective leads to higher performance across question types, question lengths, and answer lengths on the Stanford Question Answering Dataset (SQuAD) (Rajpurkar et al., 2016) compared to our DCN baseline. The improvement is especially apparent on long questions, which require the model to capture long-range dependencies between the document and the question. Our model, which we call DCN+, achieves state-of-the-art results on SQuAD, with 75.1% exact match accuracy and 83.1% F1. When ensembled, the DCN+ obtains 78.9% exact match accuracy and 86.0% F1.

## 2 DCN+

We consider the question answering task in which we are given a document and a question, and are asked to find the answer in the document. Our model is based on the DCN by Xiong et al. (2017), which consists of a coattention encoder and a dynamic decoder. The encoder first encodes the question and the document separately, then builds a codependent representation through coattention. The decoder then produces a start and end point estimate given the coattention. The DCN decoder is dynamic in the sense that it iteratively estimates the start and end positions, stopping when estimates between iterations converge to the same positions or when a predefined maximum number of iterations is reached. We make two significant changes to the DCN by introducing a deep residual coattention encoder and a mixed training objective that combines cross entropy loss from maximum likelihood estimation and reinforcement learning rewards from self-critical policy learning.

### 2.1 DEEP RESIDUAL COATTENTION ENCODER

Because it only has a single-layer coattention encoder, the DCN is limited in its ability to compose complex input representations. Vaswani et al. (2017) proposed stacked self-attention modules to facilitate signal traversal. They also showed that the network's ability to model long-range dependencies can be improved by reducing the length of signal paths. We propose two modifications to the coattention encoder to leverage these findings. First, we extend the coattention encoder with self-attention by stacking coattention layers. This allows the network to build richer representations over the input. Second, we merge coattention outputs from each layer with residual connections. This reduces the length of signal paths. Our encoder is shown in Figure 1.

Suppose we are given a document of $m$ words and a question of $n$ words. Let $L^D \in \mathbb{R}^{e \times m}$ and $L^Q \in \mathbb{R}^{e \times n}$ respectively denote the word embeddings for the document and the question, where $e$ is the dimension of the word embeddings. We obtain document encodings $E_1^D$ and question

encodings $E_1^Q$ through a bidirectional Long Short-Term Memory Network (LSTM) (Hochreiter & Schmidhuber, 1997), where we use integer subscripts to denote the coattention layer number.

$$E_1^D = \text{biLSTM}_1\left(L^D\right) \in \mathbb{R}^{h\times(m+1)} \tag{1}$$

$$E_1^Q = \tanh\left(W\ \text{biLSTM}_1\left(L^Q\right) + b\right) \in \mathbb{R}^{h\times(n+1)} \tag{2}$$

Here, $h$ denotes the hidden state size and the $+1$ indicates the presence of an additional sentinel word which allows the coattention to not focus on any part of the input. Like the original DCN, we add a non-linear transform to the question encoding.

We compute the affinity matrix between the document and the question as $A = \left(E_1^D\right)^\mathsf{T} E_1^Q \in \mathbb{R}^{(m+1)\times(n+1)}$. Let $\text{softmax}\left(X\right)$ denote the softmax operation over the matrix $X$ that normalizes $X$ column-wise. The document summary vectors and question summary vectors are computed as

$$S_1^D = E_1^Q\, \text{softmax}\left(A^\mathsf{T}\right) \in \mathbb{R}^{h\times(m+1)} \tag{3}$$

$$S_1^Q = E_1^D\, \text{softmax}\left(A\right) \in \mathbb{R}^{h\times(n+1)} \tag{4}$$

We define the document coattention context as follows. Note that we drop the dimension corresponding to the sentinel vector – it has already been used during the summary computation and is not a potential position candidate for the decoder.

$$C_1^D = S_1^Q\, \text{softmax}\left(A^\mathsf{T}\right) \in \mathbb{R}^{h\times m} \tag{5}$$

We further encode the summaries using another bidirectional LSTM.

$$E_2^D = \text{biLSTM}_2\left(S_1^D\right) \in \mathbb{R}^{2h\times m} \tag{6}$$

$$E_2^Q = \text{biLSTM}_2\left(S_1^Q\right) \in \mathbb{R}^{2h\times n} \tag{7}$$

Equation 3 to equation 5 describe a single coattention layer. We compute the second coattention layer in a similar fashion. Namely, let $\text{coattn}$ denote a multi-valued mapping whose inputs are the two input sequences $E_1^D$ and $E_1^Q$. We have

$$\text{coattn}_1\left(E_1^D, E_1^Q\right) \rightarrow S_1^D, S_1^Q, C_1^D \tag{8}$$

$$\text{coattn}_2\left(E_2^D, E_2^Q\right) \rightarrow S_2^D, S_2^Q, C_2^D \tag{9}$$

The output of our encoder is then obtained as

$$U = \text{biLSTM}\left(\text{concat}\left(E_1^D; E_2^D; S_1^D; S_2^D; C_1^D; C_2^D\right)\right) \in \mathbb{R}^{2h\times m} \tag{10}$$

where $\text{concat}\left(A, B\right)$ denotes the concatenation between the matrices $A$ and $B$ along the first dimension.

This encoder is different than the original DCN in its depth and its use of residual connections. We use not only the output of the deep coattention network $C_2^D$ as input to the final bidirectional LSTM, but add skip connections to initial encodings $E_1^D$, $E_2^D$, summary vectors $S_1^D$, $S_2^D$, and coattention context $C_1^D$. This is akin to transformer networks (Vaswani et al., 2017), which achieved state-of-the-art results on machine translation using deep self-attention layers to help model long-range dependencies, and residual networks (He et al., 2016), which achieved state-of-the-art results in image classification through the addition of skip layer connections to facilitate signal propagation and alleviate gradient degradation.

## 2.2 Mixed objective using self-critical policy learning

The DCN produces a distribution over the start position of the answer and a distribution over the end position of the answer. Let $s$ and $e$ denote the respective start and end points of the ground truth answer. Because the decoder of the DCN is dynamic, we denote the start and end distributions produced at the $t$th decoding step by $p_t^{\text{start}} \in \mathbb{R}^m$ and $p_t^{\text{end}} \in \mathbb{R}^m$. For convenience, we denote the greedy estimate of the start and end positions by the model at the $t$th decoding step by $s_t$ and $e_t$. Moreover, let $\Theta$ denote the parameters of the model.

Similar to other question answering models, the DCN is supervised using the cross entropy loss on the start position distribution and the end position distribution:

$$l_{ce}(\Theta) = -\sum_t \left( \log p_t^{\text{start}} \left( s \mid s_{t-1}, e_{t-1}; \Theta \right) + \log p_t^{\text{end}} \left( e \mid s_{t-1}, e_{t-1}; \Theta \right) \right) \tag{11}$$

Equation 11 states that the model accumulates a cross entropy loss over each position during each decoding step given previous estimates of the start and end positions.

The question answering task consists of two evaluation metrics. The first, exact match, is a binary score that denotes whether the answer span produced by the model has exact string match with the ground truth answer span. The second, F1, computes the degree of word overlap between the answer span produced by the model and the ground truth answer span. We note that there is a disconnect between the cross entropy optimization objective and the evaluation metrics. For example, suppose we are given the answer estimates $A$ and $B$, neither of which match the ground truth positions. However, $A$ has an exact *string* match with the ground truth answer whereas $B$ does not. The cross entropy objective penalizes $A$ and $B$ equally, despite the former being correct under both evaluation metrics. In the less extreme case where $A$ does not have exact match but has some degree of word overlap with the ground truth, the F1 metric still prefers $A$ over $B$ despite its wrongly predicted positions.

We encode this preference using reinforcement learning, using the F1 score as the reward function. Let $\hat{s}_t \sim p_t^{\text{start}}$ and $\hat{e}_t \sim p_t^{\text{start}}$ denote the sampled start and end positions from the estimated distributions at decoding step $t$. We define a trajectory $\hat{\tau}$ as a sequence of sampled start and end points $\hat{s}_t$ and $\hat{e}_t$ through all $T$ decoder time steps. The reinforcement learning objective is then the negative expected rewards $R$ over trajectories.

$$
\begin{aligned}
l_{rl}(\Theta) &= -\mathbb{E}_{\hat{\tau} \sim p_\tau} \left[ R\left( s, e, \hat{s}_T, \hat{e}_T; \Theta \right) \right] \tag{12} \\
&\approx -\mathbb{E}_{\hat{\tau} \sim p_\tau} \left[ F_1 \left( \text{ans} \left( \hat{s}_T, \hat{e}_T \right), \text{ans} \left( s, e \right) \right) - F_1 \left( \text{ans} \left( s_T, e_T \right), \text{ans} \left( s, e \right) \right) \right] \tag{13}
\end{aligned}
$$

We use $F_1$ to denote the F1 scoring function and $\text{ans}(s, e)$ to denote the answer span retrieved using the start point $s$ and end point $e$. In equation 13, instead of using only the F1 word overlap as the reward, we subtract from it a baseline. Greensmith et al. (2001) show that a good baseline reduces the variance of gradient estimates and facilitates convergence. In our case, we employ a self-critic (Konda & Tsitsiklis, 1999) that uses the F1 score produced by the current model during greedy inference without teacher forcing.

For ease of notation, we abbreviate $R\left( s, e, \hat{s}_T, \hat{e}_T; \Theta \right)$ as $R$. As per Sutton et al. (1999) and Schulman et al. (2015), the expected gradient of a non-differentiable reward function can be computed as

$$
\begin{aligned}
\nabla_\Theta l_{rl}(\Theta) &= -\nabla_\Theta \left( \mathbb{E}_{\hat{\tau} \sim p_\tau} [R] \right) \tag{14} \\
&= -\mathbb{E}_{\hat{\tau} \sim p_\tau} \left[ R \nabla_\Theta \log p_\tau (\tau; \Theta) \right] \tag{15} \\
&= -\mathbb{E}_{\hat{\tau} \sim p_\tau} \left[ R \nabla_\Theta \left( \sum_t^T \left( \log p_t^{\text{start}} \left( \hat{s}_t | \hat{s}_{t-1}, \hat{e}_{t-1}; \Theta \right) + \log p_t^{\text{end}} \left( \hat{e}_t | \hat{s}_{t-1}, \hat{e}_{t-1}; \Theta \right) \right) \right) \right] \\
&\approx -R \nabla_\Theta \left( \sum_t^T \left( \log p_t^{\text{start}} \left( \hat{s}_t | \hat{s}_{t-1}, \hat{e}_{t-1}; \Theta \right) + \log p_t^{\text{end}} \left( \hat{e}_t | \hat{s}_{t-1}, \hat{e}_{t-1}; \Theta \right) \right) \right) \tag{16}
\end{aligned}
$$

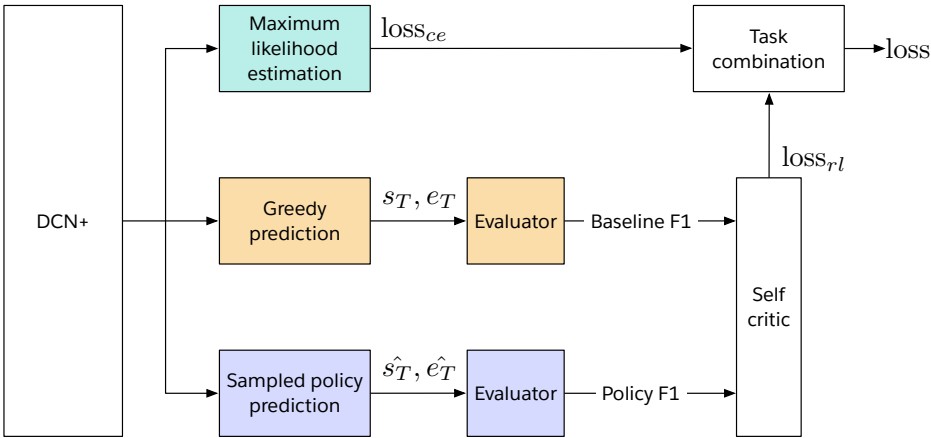

Figure 2: Computation of the mixed objective.

In equation 16, we approximate the expected gradient using a single Monte-Carlo sample $\tau$ drawn from $p_\tau$. This sample trajectory $\tau$ contains the start and end positions $\hat{s}_t$ and $\hat{e}_t$ sampled during all decoding steps.

One of the key problems in applying RL to natural language processing is the discontinuous and discrete space the agent must explore in order to find a good policy. For problems with large exploration space, RL approaches tend to be applied as fine-tuning steps after a maximum likelihood model has already been trained (Paulus et al., 2017; Wu et al., 2016). The resulting model is constrained in its exploration during fine-tuning because it is biased by heavy pretraining. We instead treat the optimization problem as a multi-task learning problem. The first task is to optimize for positional match with the ground truth answer using the the cross entropy objective. The second task is to optimize for word overlap with the ground truth answer with the self-critical reinforcement learning objective. In a similar fashion to Kendall et al. (2017), we combine the two losses using homoscedastic uncertainty as task-dependent weightings.

$$l = \frac{1}{2\sigma_{ce}^2} l_{ce}\left(\Theta\right) + \frac{1}{2\sigma_{rl}^2} l_{rl}\left(\Theta\right) + \log \sigma_{ce}^2 + \log \sigma_{rl}^2 \qquad (17)$$

Here, $\sigma_{ce}$ and $\sigma_{rl}$ are learned parameters. The gradient of the cross entropy objective can be derived using straight-forward backpropagation. The gradient of the self-critical reinforcement learning objective is shown in equation 16. Figure 2 illustrates how the mixed objective is computed. In practice, we find that adding the cross entropy task significantly facilitates policy learning by pruning the space of candidate trajectories - without the former, it is very difficult for policy learning to converge due to the large space of potential answers, documents, and questions.

## 3  EXPERIMENTS

We train and evaluate our model on the Stanford Question Answering Dataset (SQuAD). We show our test performance of our model against other published models, and demonstrate the importance of our proposals via ablation studies on the development set. To preprocess the corpus, we use the reversible tokenizer from Stanford CoreNLP (Manning et al., 2014). For word embeddings, we use GloVe embeddings pretrained on the 840B Common Crawl corpus (Pennington et al., 2014) as well as character ngram embeddings by Hashimoto et al. (2017). In addition, we concatenate these embeddings with context vectors (CoVe) trained on WMT (McCann et al., 2017). For out of vocabulary words, we set the embeddings and context vectors to zero. We perform word dropout on the document which zeros a word embedding with probability 0.075. In addition, we swap the first maxout layer of the highway maxout network in the DCN decoder with a sparse mixture of experts layer (Shazeer et al., 2017). This layer is similar to the maxout layer, except instead of taking the top scoring expert, we take the top $k = 2$ expert. The model is trained using ADAM (Kingma & Ba,

2014) with default hyperparameters. Hyperparameters of our model are identical to the DCN. We implement our model using PyTorch.

## 3.1 RESULTS

| Model | Single Model Dev | | Single Model Test | | Ensemble Test | |
|---|---|---|---|---|---|---|
| | EM | F1 | EM | F1 | EM | F1 |
| DCN+ (ours) | **74.5%** | **83.1%** | **75.1%** | **83.1%** | **78.9%** | **86.0%** |
| rnet | 72.3% | 80.6% | 72.3% | 80.7% | 76.9% | 84.0% |
| DCN w/ CoVe (baseline) | 71.3% | 79.9% | – | – | – | – |
| Mnemonic Reader | 70.1% | 79.6% | 69.9% | 79.2% | 73.7% | 81.7% |
| Document Reader | 69.5% | 78.8% | 70.0% | 79.0% | – | – |
| FastQA | 70.3% | 78.5% | 70.8% | 78.9% | – | – |
| ReasoNet | – | – | 69.1% | 78.9% | 73.4% | 81.8% |
| SEDT | 67.9% | 77.4% | 68.5% | 78.0% | 73.0% | 80.8% |
| BiDAF | 67.7% | 77.3% | 68.0% | 77.3% | 73.7% | 81.5% |
| DCN | 65.4% | 75.6% | 66.2% | 75.9% | 71.6% | 80.4% |

Table 1: Test performance on SQuAD. The papers are as follows: rnet (Microsoft Asia Natural Language Computing Group, 2017), SEDT (Liu et al., 2017), BiDAF (Seo et al., 2017), DCN w/ CoVe (McCann et al., 2017), ReasoNet (Shen et al., 2017), Document Reader (Chen et al., 2017), FastQA (Weissenborn et al., 2017), DCN (Xiong et al., 2017). The CoVe authors did not submit their model, which we use as our baseline, for SQuAD test evaluation.

The performance of our model is shown in Table 1. Our model achieves state-of-the-art results on SQuAD dataset with 75.1% exact match accuracy and 83.1% F1. When ensembled, our model obtains 78.9% exact match accuracy and 86.0% F1. To illustrate the effectiveness of our proposals, we use the DCN with context vectors as a baseline (McCann et al., 2017). This model is identical to the DCN by Xiong et al. (2017), except that it augments the word representations with context vectors trained on WMT16.

**Comparison to baseline DCN with CoVe.**    DCN+ outperforms the baseline by 3.2% exact match accuracy and 3.2% F1 on the SQuAD development set. Figure 3 shows the consistent performance gain of DCN+ over the baseline across question types, question lengths, and answer lengths. In particular, DCN+ provides a significant advantage for long questions.

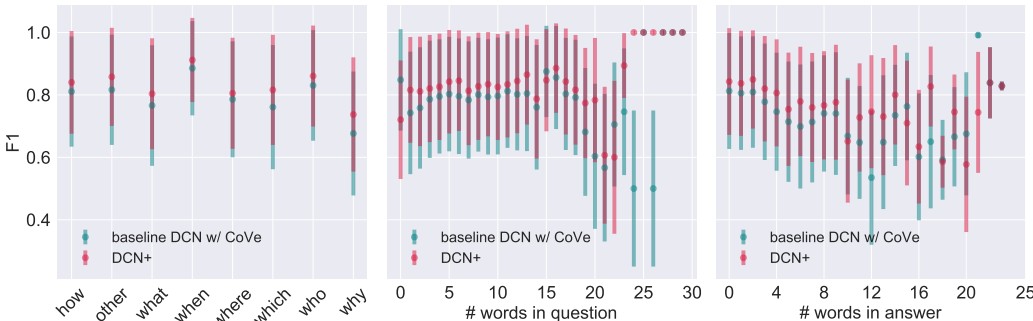

Figure 3: Performance comparison between DCN+ and the baseline DCN with CoVe on the SQuAD development set.

**Ablation study.**    The contributions of each part of our model are shown in Table 2. We note that the deep residual coattention yielded the highest contribution to model performance, followed by the mixed objective. The sparse mixture of experts layer in the decoder added minor improvements to the model performance.

| Model | EM | ΔEM | F1 | ΔF1 |
|---|---|---|---|---|
| DCN+ (ours) | 74.5% | – | 83.1% | – |
| - Deep residual coattention | 73.1% | -1.4% | 81.5% | -1.6% |
| - Mixed objective | 73.8% | -0.7% | 82.1% | -1.0% |
| - Mixture of experts | 74.0% | -0.5% | 82.4% | -0.7% |
| DCN w/ CoVe (baseline) | 71.3% | -3.2% | 79.9% | -3.2% |

Table 2: Ablation study on the development set of SQuAD.

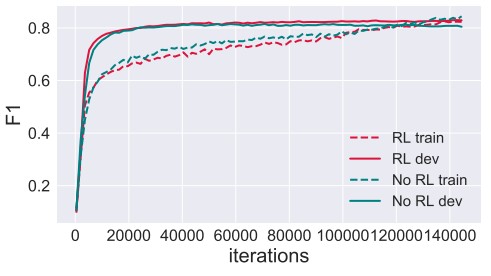

(a) Entirety of the training curve.

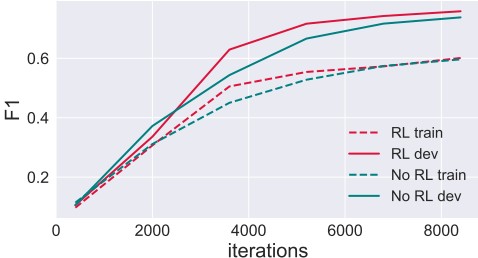

(b) A closeup of the early stages of training.

Figure 4: Training curve of DCN+ with and without reinforcement learning. In the latter case, only the cross entropy objective is used. The mixed objective initially performs worse as it begins policy learning from scratch, but quickly outperforms the cross entropy model.

**Mixed objective convergence.** The training curves for DCN+ with reinforcement learning and DCN+ without reinforcement learning are shown in Figure 4 to illustrate the effectiveness of our proposed mixed objective. In particular, we note that without mixing in the cross entropy loss, it is extremely difficult to learn the policy. When we combine the cross entropy loss with the reinforcement learning objective, we find that the model initially performs worse early on as it begins policy learning from scratch (shown in Figure 4b). However, with the addition of cross entropy loss, the model quickly learns a reasonable policy and subsequently outperforms the purely cross entropy model (shown in Figure 4a).

**Sample predictions.** Figure 5 compares predictions by DCN+ and by the baseline on the development set. Both models retrieve answers that have sensible entity types. For example, the second example asks for "what game" and both models retrieve an American football game; the third example asks for "type of Turing machine" and both models retrieve a type of turing machine. We find, however, that DCN+ consistently make less mistakes on finding the correct entity. This is especially apparent in the examples we show, which contain several entities or candidate answers of the correct type. In the first example, Gasquet wrote about the plague and called it "Great Pestilence". While he likely did think of the plague as a "great pestilence", the phrase "suggested that it would appear to be some form of ordinary Eastern or bubonic plague" provides evidence for the correct answer – "some form of ordinary Eastern or bubonic plague". Similarly, the second example states that Thomas Davis was injured in the "NFC Championship Game", but the game he insisted on playing in is the "Super Bowl". Finally, "multi-tape" and "single-tape" both appear in the sentence that provides provenance for the answer to the question. However, it is the "single-tape" Turing machine that implies quadratic time. In these examples, DCN+ finds the correct entity out of ones that have the right type whereas the baseline does not.

## 4 RELATED WORK

**Neural models for question answering.** Current state-of-the-art approaches for question answering over unstructured text tend to be neural approaches. Wang & Jiang (2017) proposed one of the first conditional attention mechanisms in the Match-LSTM encoder. Coattention (Xiong et al., 2017), bidirectional attention flow (Seo et al., 2017), and self-matching attention (Microsoft Asia Natural Language Computing Group, 2017) all build codependent representations of the question and the document. These approaches of conditionally encoding two sequences are widely used in

What did Gasquet think the plague was?

The historian Francis Aidan Gasquet wrote about the 'Great Pestilence' in 1893 and suggested that "it would appear to be some form of the ordinary Eastern or bubonic plague". He was able to adopt the epidemiology of the bubonic plague for the Black Death for the second edition in 1908, implicating rats and fleas in the process, and his interpretation was widely accepted for other ancient and medieval epidemics, such as the Justinian plague that was prevalent in the Eastern Roman Empire from 541 to 700 CE.

---

What game did Thomas Davis say he would play in, despite breaking a bone earlier on?

Carolina suffered a major setback when Thomas Davis, an 11-year veteran who had already overcome three ACL tears in his career, went down with a broken arm in the NFC Championship Game. Despite this, he insisted he would still find a way to play in the Super Bowl. His prediction turned out to be accurate.

---

A language solved in quadratic time implies the use of what type of Turing machine?

But bounding the computation time above by some concrete function f(n) often yields complexity classes that depend on the chosen machine model. For instance, the language {xx | x is any binary string} can be solved in linear time on a multi-tape Turing machine, but necessarily requires quadratic time in the model of single-tape Turing machines. If we allow polynomial variations in running time, Cobham-Edmonds thesis states that "the time complexities in any two reasonable and general models of computation are polynomially related" (Goldreich 2008, Chapter 1.2). This forms the basis for the complexity class P, which is the set of decision problems solvable by a deterministic Turing machine within polynomial time. The corresponding set of function problems is FP.

Figure 5: Predictions by DCN+ (red) and DCN with CoVe (blue) on the SQuAD development set.

question answering. After building codependent encodings, most models predict the answer by generating the start position and the end position corresponding to the estimated answer span. The generation process utilizes a pointer network (Vinyals et al., 2015) over the positions in the document. Xiong et al. (2017) also introduced the dynamic decoder, which iteratively proposes answers by alternating between start position and end position estimates, and in some cases is able to recover from initially erroneous predictions.

**Neural attention models.** Neural attention models saw early adoption in machine translation (Bahdanau et al., 2015) and has since become to de-facto architecture for neural machine translation models. Self-attention, or intra-attention, has been applied to language modeling, sentiment analysis, natural language inference, and abstractive text summarization (Chen et al., 2017; Paulus et al., 2017). Vaswani et al. (2017) extended this idea to a deep self-attentional network which obtained state-of-the-art results in machine translation. Coattention, which builds codependent representations of multiple inputs, has been applied to visual question answering (Lu et al., 2016). Xiong et al. (2017) introduced coattention for question answering. Bidirectional attention flow (Seo et al., 2017) and self-matching attention (Microsoft Asia Natural Language Computing Group, 2017) also build codependent representations between the question and the document.

**Reinforcement learning in NLP.** Many tasks in natural language processing have evaluation metrics that are not differentiable. Dethlefs & Cuayáhuitl (2011) proposed a hierarchical reinforcement learning technique for generating text in a simulated way-finding domain. Narasimhan et al. (2015) applied deep Q-networks to learn policies for text-based games using game rewards as feedback. Li et al. (2016) introduced a neural conversational model trained using policy gradient methods, whose reward function consisted of heuristics for ease of answering, information flow, and semantic coherence. Bahdanau et al. (2017) proposed a general actor-critic temporal-difference method for sequence prediction, performing metric optimization on language modeling and machine translation. Direct word overlap metric optimization has also been applied to summarization (Paulus et al., 2017), and machine translation (Wu et al., 2016).

## 5 CONCLUSION

We introduced DCN+, an state-of-the-art question answering model with deep residual coattention trained using a mixed objective that combines cross entropy supervision with self-critical policy learning. We showed that our proposals improve model performance across question types, question lengths, and answer lengths on the Stanford Question Answering Dataset ( SQuAD). On SQuAD, the DCN+ achieves 75.1% exact match accuracy and 83.1% F1. When ensembled, the DCN+ obtains 78.9% exact match accuracy and 86.0% F1.

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
