# OpenReview forum: "DCN+: Mixed Objective And Deep Residual Coattention for Question Answering"
_ICLR.cc/2018/Conference — Accept (Poster)_

### Official Review · AnonReviewer2 · 2017-11-25

**Rating:** 7
**Confidence:** 4

**Review:**

Summary:
This paper proposed an extension of the dynamic coattention network (DCN) with deeper residual layers and self-attention. It also introduced a mixed objective with self-critical policy learning to encourage predictions with high word overlap with the gold answer span. The resulting DCN+ model achieved significant improvement over DCN.

Strengths:
The model and the mixed objective is well-motivated and clearly explained.
Near state-of-the-art performance on SQuAD dataset (according to the SQuAD leaderboard).

Other questions and comments:
The ablation shows 0.7 improvement on EM with mixed objective. It is interesting that the mixed objective (which targets F1) also brings improvement on EM.

---

> ### Author Response · Authors · 2017-12-30
> **RE: Review**
>
> Thanks for your comments. In practice the F1 and EM metrics are closely correlated. We chose to use the F1 score as a metric because it offers fine grain signals as to how well the span predicted matches the ground truth span, whereas the EM score only rewards exact gloss matches.

---

### Official Review · AnonReviewer3 · 2017-11-28
**An improvement of DCN model**

**Rating:** 6
**Confidence:** 4

**Review:**

This paper proposed an improved version of dynamic coattention networks, which is used for question answering tasks. Specifically, there are 2 aspects to improve DCN: one is to use a mixed objective that combines cross entropy with self-critical policy learning, the other one is to imporve DCN with deep residual coattention encoder. The proposed model achieved STOA performance on Stanford Question Asnwering Dataset and several ablation experiments show the effectiveness of these two improvements. Although DCN+ is an improvement of DCN, I think the improvement is not incremental.

One question is that since the model is compicated, will the authors release the source code to repeat all the experimental results?

---

> ### Author Response · Authors · 2017-12-30
> **RE: An improvement of DCN model**
>
> Thanks for your comments. We can try to release the source code after the decision process.

---

### Official Review · AnonReviewer1 · 2017-11-30
**Significant improvement of DCN answer selection models using mixed objectives and 2 stacked levels of coattention**

**Rating:** 8
**Confidence:** 2

**Review:**

The authors of this paper propose some extensions to the Dynamic Coattention Networks models presented last year at ICLR. First they modify the architecture of the answer selection model by adding an extra coattention layer to improve the capture of dependencies between question and answer descriptions. The other main modification is to train their DCN+ model using both cross entropy loss and F1 score (using RL supervision) in order to  reward the system for making partial matching predictions. Empirical evaluations conducted on the SQuAD dataset indicates that this architecture achieves an improvement of at least 3%, both on F1 and exact match accuracy, over other comparable systems. An ablation study clearly shows the contribution of the deep coattention mechanism and mixed objective training on the model performance.

The paper is well written, ideas are presented clearly and the experiments section provide interesting insights such as the impact of RL on system training or the capability of the model to handle long questions and/or answers. It seems to me that this paper is a significant contribution to the field of question answering systems.

---

> ### Author Response · Authors · 2017-12-30
> **RE: Significant improvement of DCN answer selection models using mixed objectives and 2 stacked levels of coattention**
>
> Thank you for your comments!

---

### Author Response · Authors · 2017-11-10
**Errata**

In Equation 17 (page 5), we made a typo in that we did not include the regularization terms $$\log \sigma_{ce}^2 + \log \sigma_{rl}^2$$.

---

### Public Comment · (anonymous) · 2017-12-03
**Only SQuAD Evaluation!?**

I noticed that you only evaluate against SQuAD which is known to be a bad dataset for evaluating machine comprehension. It has only short documents and most of the answers are easily extractable. This is a bit troubling especially given that there are plenty of good and much more complex datasets out there, e.g., TriviaQA, NewsQA, just to mention a few. It feels like we are totally overfitting on a simple dataset. Would it be possible to also provide results on one of those, otherwise it is really hard to judge whether there is indeed any significant improvement. I think this is a big issue.

---

> ### Author Response · Authors · 2017-12-04
> **RE: SQuAD evaluation**
>
> Hi,
>
> You are correct in that we only evaluate on the SQuAD dataset. In short, we agree with your sentiment that it would be interesting to evaluate on other datasets, however we respectfully disagree that SQuAD is known to be a bad dataset. In fact, we feel that it is one of the best datasets for reading comprehension. There seems to be agreement within the community about this in that
>
> 1. SQuAD received the best resource paper award at EMNLP
> 2. It is highly competitive, drawing significant participation from not only the authors' institution but other well-regarded academic and industry institutions (e.g. AI2, MSR (A), FAIR, CMU, Google, Stanford, Montreal, NYU ...).
> 3. It has shown very useful downstream applications and impact (e.g. http://nlp.cs.washington.edu/zeroshot/)
>
> Given these points, we feel that the performance gains afforded by our proposal (~3% F1) is significant, given that the top models on the leaderboard are within ~1% F1 of each other. We think these techniques are beneficial to the community at large.
>
> Of course, the fact that the other datasets do not (yet) have the above distinctions does not make them less interesting. We think that each dataset has its pros and cons. For example, TriviaQA is labeled via distant supervision. NewsQA  frankly has not been very popular (2 leaderboard submissions in ~1 year), and there seems to be concerns regarding its evaluation (see https://openreview.net/forum?id=ry3iBFqgl), though the authors seems to have made some enhancements since then. Given the higher popularity and competitiveness of SQuAD, we felt that it is a better choice on which to compare our proposal with the best models developed by the community.
>
> Nevertheless, the two datasets you mentioned are either larger and longer (TriviaQA) or at least longer (NewsQA) than SQuAD. We will attempt to evaluate on one of these two datasets, however given the time constraints we are unlikely to be able to fine tune the model. I will update here with results once we obtain them.

---

> > ### Public Comment · (anonymous) · 2017-12-05
> > **RE: RE: SQuAD evaluation**
> >
> > I disagree with that. SQuAD has been awarded best resource reward (which was fine one year ago), however, the dataset itself consists to a large degree of simple paraphrases and the context size is simply to small to be a good RC dataset. Various recent dataset papers have shown the limitations of SQuAD.
> >
> > The fact that well-regarded academic and industry institutions are all working on that has something to do with PR.
> >
> > Furthermore, works of [1] and  [2] show that the dataset is not challenging. Especially [1] shows that models trained on SQuAD are easily fooled. So no reading comprehension.
> >
> > I think it is really problematic that the community overfits on certain datasets even when there are better alternatives. SQuAD is a great proof of concept dataset but not much more than that today.
> >
> > The reason nobody is evaluating on the other datasets is that they are more challenging to handle because of their size and context length, not because they are 'bad'.
> >
> > [1] Robin Jia, and Percy Liang. "Adversarial examples for evaluating reading comprehension systems." EMNLP (2017).
> > [2] Dirk Weissenborn, Georg Wiese, Laura Seiffe. "Making Neural QA as Simple as Possible but not Simpler". CoNLL 2017

---

> > > ### Author Response · Authors · 2017-12-05
> > > **RE: RE: RE:**
> > >
> > > It is not our intention to characterize the other two datasets as bad. In fact, we think highly of the TriviaQA dataset and are investigating potential applications for it. We simply meant that each dataset has its own merits and drawbacks.
> > >
> > > Furthermore, it sounds like the anonymous reviewer thinks that the adversarial methods by Jia et al demonstrates the limitations of SQuAD. We disagree. We think that Robin's work rather demonstrates the limitations of state of the art reading comprehension models. In particular, we speculate that similar methods can be applied to the other datasets. Finally, to say that SQuAD models do not do reading comprehension is, in my opinion, unfair, and trivializes genuine hard work by the community.

---

> > > > ### Public Comment · (anonymous) · 2017-12-05
> > > > **RE: RE: RE: RE:**
> > > >
> > > > Just to clarify, I do not say the models cannot do reading comprehension but that models trained on SQuAD are not doing it (which is due to SQuAD not the models), hence the results by Robin et al. The problem is that simple heuristics are good enough to almost solve SQuAD.
> > > >
> > > > (I am not an anonymous reviewer, just a concerned reader)

---

### Public Comment · (anonymous) · 2017-12-04
**Ablation without CoVe? Only SoTA by using newer embeddings?**

Hi, most models this paper compares to are trained with GloVe embeddings but you only show results with CoVe (if I am not mistaken). Given the large boost of CoVe to the original model, it looks like this model is only able to achieve SotA because it uses CoVe and not because of the additional extensions. The mentioned 3% boost to the original model without CoVe would result in a much lower score which would probably **not be SoTA**.

Is this correct?

---

> ### Author Response · Authors · 2017-12-04
> **RE: Ablation without CoVe**
>
> Hi,
>
> You are correct in that CoVe does provide a significant performance gain, as demonstrated by McCann et al. (https://arxiv.org/abs/1708.00107). However, CoVe itself, when combined with the original DCN, does not obtain state of the art performance whereas this work does (please see Table 1 of our paper). In addition, we feel that the performance gain provided by deep residual coattention and mixed objective are significant (3.2% dev F1) given the competitive nature of the task. For reference on how significant a 3% F1 gain is, the top 5 state of the art models on the leaderboard are within ~1% dev F1 of each other.
>
> In addition, CoVe is focused on the encoder of the model, whereas our work focuses on the coattention and the mixed objective. Our additions are applicable to other types of encoders as well.
>
> We decided to perform ablation studies with respect to DCN with CoVe because it seemed like a natural foundation to build upon, but I agree with your sentiment that we should also evaluate our proposed additions without CoVe. We can try to perform this experiment and update here with the results.
>
> Thanks!

---

> > ### Public Comment · (anonymous) · 2017-12-05
> > **RE: RE: Ablation without CoVe**
> >
> > The point I was making is that SotA is achieved only through CoVe, otherwise your results would probably be 2-3% lower, which is not that convincing anymore. Improving upon DCN itself by 3% has been achieved by other, simpler models (e.g., Document Reader Chen et al 2017).
> >
> > In general, making a model slightly deeper is not novel but a bit of engineering. The mixed objective is a neat little contribution (however, there is the Reinforced Mnemonic Reader which does something similar already, although probably written in parallel). Anyway, you are directly optimizing your evaluation metric (which is fine) that probably learns to give better bounds for your answer selection. The problem with the bounds can be seen by the rather large gap between F1 and Exact match. This gap is less pronounced in TriviaQA for instance where the mixed objective would probably have a much smaller effect. This results in climbing the SQuAD ladder but probably not really improving reading comprehension.

---

> > > ### Author Response · Authors · 2018-01-02
> > > **RE: RE: RE: Ablation without CoVe**
> > >
> > > We have performed another ablation study in which we train our DCN+ model without CoVe. This variant obtained 81.4% F1 on the dev set of SQuAD. This number also outperforms all other models studied in the paper, which suggests that our proposed changes to the original DCN are significant. For reference, DCN+ with CoVe obtained 83.1% on the dev set, and the test numbers tend to be a bit higher.

---

### Public Comment · (anonymous) · 2018-01-20
**Training vs dev ratio**

The performance of the model on SQuAD dataset is impressive. In addition to the performance on the test set, we are also interested in the sample complexity of the proposed model. Currently, the SQuAD dataset splits the collection of passages into a training set, a development set, and a test set in a ratio of 80%:10%:10% where the test set is not released. Given the released training and dev set, we are wondering what would happen if we split the data in a different ratio, for example, 50% for training and the rest 50% for dev. We will really appreciate it if the authors can report the model performance (on training/dev respectively) under this scenario.

---

### Decision · Program_Chairs · 2018-01-29
**ICLR 2018 Conference Acceptance Decision**

**Decision:**

Accept (Poster)

**Comment:**

This is an interesting paper that provides modeling improvements over several strong baselines and presents SOTA on Squad.  One criticism of the paper is that it evaluates only on Squad, which is somewhat of an artificial task, but we think for publication purposes at ICLR, the paper has a reasonable set of components.